# Nanofunctionalization of Additively Manufactured Titanium Substrates for Surface-Enhanced Raman Spectroscopy Measurements

**DOI:** 10.3390/ma15093108

**Published:** 2022-04-25

**Authors:** Marcin Pisarek, Robert Ambroziak, Marcin Hołdyński, Agata Roguska, Anna Majchrowicz, Bartłomiej Wysocki, Andrzej Kudelski

**Affiliations:** 1Institute of Physical Chemistry, Polish Academy of Sciences, Kasprzaka 44/52, 01-224 Warsaw, Poland; rambroziak@ichf.edu.pl (R.A.); mholdynski@ichf.edu.pl (M.H.); aroguska@ichf.edu.pl (A.R.); 2Faculty of Materials Science and Engineering, Warsaw University of Technology, Wołoska 141, 02-507 Warsaw, Poland; aankam@gmail.com; 3Center of Digital Science and Technology, Cardinal Stefan Wyszynski University in Warsaw, Woycickiego 1/3, 01-938 Warsaw, Poland; b.wysocki@uksw.edu.pl; 4Faculty of Chemistry, University of Warsaw, Pasteura 1, 02-093 Warsaw, Poland; akudel@chem.uw.edu.pl

**Keywords:** additive manufacturing, powder bed fusion (PBF), CP titanium, anodic oxidation, TiO_2_ nanotubes, Ag nanoparticles, SERS platforms, plasmonic substrates

## Abstract

Powder bed fusion using a laser beam (PBF-LB) is a commonly used additive manufacturing (3D printing) process for the fabrication of various parts from pure metals and their alloys. This work shows for the first time the possibility of using PBF-LB technology for the production of 3D titanium substrates (Ti 3D) for surface-enhanced Raman scattering (SERS) measurements. Thanks to the specific development of the 3D titanium surface and its nanoscale modification by the formation of TiO_2_ nanotubes with a diameter of ~80 nm by the anodic oxidation process, very efficient SERS substrates were obtained after deposition of silver nanoparticles (0.02 mg/cm^2^, magnetron sputtering). The average SERS enhancement factor equal to 1.26 × 10^6^ was determined for pyridine (0.05 M + 0.1 M KCl), as a model adsorbate. The estimated enhancement factor is comparable with the data in the literature, and the substrate produced in this way is characterized by the high stability and repeatability of SERS measurements. The combination of the use of a printed metal substrate with nanofunctionalization opens a new path in the design of SERS substrates for applications in analytical chemistry. Methods such as SEM scanning microscopy, photoelectron spectroscopy (XPS) and X-ray diffraction analysis (XRD) were used to determine the morphology, structure and chemical composition of the fabricated materials.

## 1. Introduction

Surface-enhanced Raman spectroscopy (SERS) offers a wide range of possibilities in the study of various types of analytes on surfaces appropriately prepared for this purpose. Usually, these are nanostructured substrates with a high degree of surface development based on plasmonic metals [1]. From a historical point of view, the first substrates used in SERS spectroscopy were based on nanostructured silver obtained by electrochemical pretreatment [2]. On such surfaces, a very strong Raman signal from adsorbed pyridine was observed [1,2,3,4]. Since then, there has been a continuous search for durable, stable and repeatable substrates that generate high amplification of the recorded Raman spectra [5]. Such investigations are mainly focused on designing nanoscale platforms in terms of topography and chemical composition, where plasmonic nanoparticles such as silver, gold, copper and their alloys play a key role [1,6,7]. SERS-active substrates utilize the effect of localized surface plasmon resonance (LSPR), which occurs when an electromagnetic wave with a frequency tuned to the frequency of the localized surface plasmons interacts with plasmonic nanoparticles [8,9]. For molecules in close proximity to such substrates, this effect leads to an increase in the cross-section of scattering and absorption, as well as causing the appearance of strong electromagnetic fields around the plasmonic nanoparticles (local field enhancement (LFE)) [9,10,11,12]. The magnitude of the induced electromagnetic field depends on the size, shape and distribution of the plasmonic nanoparticles (possible plasmonic couplings) [5,9,13,14]. Therefore, when plasmonic nanostructures are deposited on nonplamonic substrates, the achievable SERS enhancement factors strongly depend on the surface topography and physical properties of semiconductors [10,15]. These make it possible to control the resonance at the nanoscale and adapt it to the wavelengths to be used in the planned applications [9,10,14]. All these factors are extremely important when designing new platforms for SERS investigations. Therefore, the aim of this study was to use for the first time the powder bed fusion process to fabricate a macrorough titanium surface, which could be later functionalized by anodic oxidation to create a nanoporous surface in the form of nanotubes [16,17,18,19,20]. The nanostructure prepared in this way was covered with a deposit of silver nanoparticles by the magnetron sputtering technique (0.02 mg/cm^2^). The use of the powder bed fusion process made it possible to generate an original geometry of the SERS substrate in the submicron scale, which was further functionalized at the nanoscale. This solution represents a new proposed strategy for the design of active SERS substrates. According to our knowledge, this approach to designing active SERS platforms is an innovative solution. Until now, printed substrates based on polymers with metals were used for this purpose [21], which were then functionalized by applying, for example, a layer of silver [21]. The electroplating technique, which produced metallic layers with controlled thickness and surface morphology, turned out to be a particularly effective method in this respect [22]. The advantage of printed substrates is that the geometry of the sample surface can be freely shaped on the submicron scale [23]. Such surface geometry, however, does not meet the criterion of applicability in SERS spectroscopy, because the phenomenon of Raman signal amplification occurs most effectively at the nanoscale [11]. Therefore, further functionalization of the printed substrates is required. Typically, for this purpose, various types of coatings containing nanoparticles of plasmonic metals are used [24]. An alternative way to obtain this type of substrate can be via lithographic methods [6,25], but they are much more difficult to apply and more expensive to use. Nevertheless, their advantage is the fact that the substrates are designed and manufactured on the nanoscale from start to finish (the horizontal resolution of the method is much better than that of standard 3D printing) [6,25]. The idea of using the PBF process seems in this case an interesting solution for designing active SERS platforms.

## 2. Materials and Methods

### 2.1. Fabrication of 3D Titanium Substrates by Powder Bed Fusion Process Using a Laser Beam

Samples with a disc shape (diameter of about 14 mm and a thickness of about 0.25 mm) were fabricated by powder bed fusion technology using the Realizer SLM50 (Realizer GmbH, Borchen, Germany) selective laser melting machine. Spherical, gas-atomized and contamination-free CP titanium Grade 1 metallic powder (ECKART TLS, Bitterfeld-Wolfen, Germany) was used for the background of SERS substrates. According to the manufacturer data, the CP Ti Gr 1 powder had a diameter below 45 µm and met the American Society for Testing and Materials (ASTM, West Conshohocken, PN, USA) titanium Grade 1 requirements, and its purity was minimally 99.5 wt.% (max. 0.20 Fe, max. 0.08% C, max. 0.03% N, max. 0.015% H, max. 0.18 O, balance Ti). The substrates were fabricated using laser power of 43 W, summarized scanning speed of 325 mm/s (40 µs exposure time and a 20 µm point distance between each next point) and layer thickness set at 25 µm. The total energy density was 130 J/mm^3^. The laser scanning strategy for the individual disc alternated with a 45° rotation on each layer, while the distance between each laser scanning vector (hatch distance) was 40 µm. The support structure for the disc was fabricated using a lower laser power (30 W) but using the same laser exposure time (40 µs) in each random point of the support structure’s cross-section [10], as shown in Figure 1.

### 2.2. Formation of TiO_2_ Nanotubes on Ti Substrates Produced by PBF-LB Process

TiO_2_ NTs were fabricated in one-step anodic oxidation of 3D Ti substrates (Ti 3D). The anodization of the titanium substrates was performed in an optimized electrolyte: a glycerol/water mixture (volume ratio 50:50) with 0.27 M NH_4_F at a constant voltage of 20 V and a time of 2 h using a two-electrode system (anode: -Ti, cathode: Pt) [26,27]. After anodization, the samples were cleaned with deionized water through long-term rinsing (24 h) and subsequently dried in air. Such a procedure leads to the cleaning of the sample surface from some organic contaminants coming from the electrolyte. Thermal annealing in air was performed at 450 °C for 3 h in order to transform the TiO_2_ NTs structure from amorphous to crystalline structure [26,28,29,30] and remove the rest of the contaminants from the surfaces of the samples.

### 2.3. Deposition of Metal Nanoparticles

The structures obtained were covered with Ag (0.02 mg/cm^2^) by the DC magnetron sputtering technique using a Leica EM MED020 apparatus (Leica Microsystems GmbH, Wetzlar, Germany). The average amount of metal deposited per cm^2^ was strictly controlled by a quartz microbalance in situ (Leica EM QSG100, Leica Microsystems GmbH, Wetzlar, Germany). The configuration of the setup was perpendicular to the surface of the sample. This configuration was the most suitable for the silver to penetrate into the TiO_2_ nanopores.

### 2.4. Characterization

The surface morphology and chemical composition of the samples were examined using a scanning electron microscope (SEM, an FEI Nova NanoSEM 450, Brno, Czech Republic). For typical imaging, low-energy electron detectors, an Everhart–Thornley detector (ETD) and a through-the-lens (TLD) detector, were used, in the low- and high-resolution modes, respectively. All modes were performed in the same configuration, at a primary beam energy of 10 kV.

The chemical states of individual elements were verified by X-ray photoelectron spectroscopy (XPS) using a Microlab 350 (Thermo Electron, East Grinstead, UK) spectrometer. For this purpose, the X-ray excitation source (AlKα anode: power 300 W, voltage 15 kV, beam current 20 mA) was used. The lateral resolution of XPS analysis was about 0.2 cm^2^. The high-resolution XPS spectra were recorded using the following parameters: pass energy 40 eV, energy step size 0.1 eV. A smart function of background subtraction was used to obtain the XPS signal intensity. All the collected XPS peaks were fitted using an asymmetric Gaussian/Lorentzian mixed function. The measured binding energies were corrected in reference to the energy of C 1s at 285.0 eV. Avantage-based data system software (Version 5.9911, Thermo Fisher Scientific, Waltham, MA, USA) was used to process the data.

X-ray powder diffraction data were collected on a PANalytical Empyrean (Malvern Panalytical Ltd., Malvern, UK) diffractometer fitted with a X’Celerator detector using Ni-filtered Cu Kα radiation (λ_1_ = 1.54056 Å and λ_2_ = 1.54439 Å). Data were collected on a flat plate with θ/θ geometry on a spinning sample holder. All presented data were collected in the 2θ range 10–90°, in intervals of 0.0167°, with a scan time of 30 s per interval.

The SERS Raman spectra were measured with a Horiba Jobin-Yvon Labram HR800 spectrometer (Longjumeau, France) equipped with a Peltier-cooled CCD detector (1024 × 256 pixels), a 600 groove/mm holographic grating and an Olympus BX40 microscope (Tokyo, Japan) with a long-distance 50× objective. All Raman spectra were recorded using a diode-pumped, frequency-doubled Nd:YAG laser (532 nm). A 200 µL volume of 0.05 M pyridine in 0.1 M KCl was applied to the substrate, and the measurement itself was performed before being drop-dried. For each substrate, 400 spectra were collected from a square surface with an edge length of 50 µm.

## 3. Results and Discussion

Figure 2 shows the topography of the 3D titanium substrates (Ti 3D) fabricated by the powder bed fusion process using a laser beam. Microscopic observations revealed a large development of the surface in the form of spherical growths, which are unmelted fully titanium powders occurring typically after the PBF process [31,32]. This phenomenon is explained by the nature of the melt pool during fabrication. The high-energy laser beam produces sufficient heat to create the melt pool at the surface of the powder bed [33]. The temperature surrounding the melt pool is immediately increased by heat conduction. Some of the powder particles, regardless of the CAD design, placed in the heat conduction zone do not melt but may become lightly sintered to the surface of the part [34,35]. In our study, the unmelted powder particles on the disc’s surfaces are also caused by heat transfer, between the melt pool and unmelted powder in the bed, affecting their sintering with the surface [36]. The size of unmelted fully powder particles is within the range of the used raw powders and is from a few to several dozen micrometers. Slightly sintered titanium particles remain on the surface even after washing in an ultrasonic cleaner. As a result, an interesting topography of the sample surface was created for the SERS applications. Nevertheless, the obtained microscale morphology may not meet the key requirements for the design of active SES substrates after deposition of a plasmonic metal such as Ag. Thus, such a surface generates some irregularities that may have an influence on the variation of the intensity of the measured SERS spectra, knowing that the phenomenon of localized resonance of surface plasmons occurs most effectively in smaller nanostructures [37,38].

Therefore, considering the above criterion, the as-made laser powder bed fused 3D titanium substrates underwent further surface functionalization. Figure 3 shows the SEM images of the 3D titanium substrate surface morphology after anodic oxidation in an electrolyte based on glycerin and water with the addition of ammonium fluoride at a voltage of 20 V. This type of treatment led to the formation of titanium oxide nanotubes, which are well visible. The nanotubes accurately reflected the state of the initial surface, thanks to which development of the 3D titanium surface at the nanoscale was obtained, as shown in Figure 3. The nanotubes evenly covered both the spaces between the spherical accretions and the accretions themselves. The size of the nanotubes formed at 20 V was around 80 nm, which is in line with our previous research in this field [39]. The size of nanotubes can be freely changed depending on the value of the applied voltage of the anodic oxidation process. As the voltage increases, the size of the nanotubes grows larger and larger depending on the electrolyte used [17,26,29,40].

The samples with nanotubes were then heated to 450 °C in order to transform the structure of titanium oxide from an amorphous form directly after anodization to a crystalline form, anatase, as shown in Figure 4 [26,28,29,30,37,39,41]. The XRD spectra show characteristic peaks from metallic Ti (3D-printed substrate) and from the titanium oxide phase in the form of anatase (3D-printed substrate after annealing at 450 °C). Typically, anatase occurs above 300 °C, which is consistent with the observations of other researchers for this type of system [26,40]. In addition, the heat treatment procedure mechanically stabilized the obtained nanostructure on the 3D titanium surface, because a barrier layer is formed at the oxide/metal interface, as we have shown in our previous work [41,42]. Annealing also leads to almost complete loss of the fluorides at around 300 °C and reduced other surface contamination such as hydrocarbons coming from anodic oxidation process [29]. Moreover, based on our earlier research, we know that annealing at 450 °C does not lead to any significant changes in the size of the nanotubes [39].

Our studies and also other research groups showed that heated TiO_2_ nanotubes coated with Ag, Au and Cu plasmonic nanoparticles turned out to be extremely active substrates for basic SERS research [17,18,43]. The nanotubes with this type of deposit acted as nanoresonators, amplifying the electromagnetic field at nanovolumes under the influence of the selected laser light with a wavelength close to the energy of the plasmonic resonance of the nanomaterials [17,18,20,43]. The resulting plasmonic nanostructures were characterized by a regular, highly ordered surface development, where the distances between the nanotubes of metal deposit and the resulting gaps favored the formation of “hot spots”, which were responsible for the enhancement of the SERS signal of the adsorbed probe molecule [9]. This was in accordance with electromagnetic theory, which states that the vibrations of the molecules normal to the surface of the plasmonic nanostructure are the most strongly enhanced, and so the best morphology for the occurrence of surface plasma resonance is small particles (<100 nm) with an atomically uneven surface [7,9,44,45]. In addition to the electromagnetic mechanism (EM) of the enhancement of the intensity of the Raman spectra, for this type of substrate, a chemical (so-called charge transfer (CT)) effect of the enhancement of the SERS spectra may also appear. This mechanism is associated with molecular orbital interaction between the analyte and the metal nanoparticles (NPs) deposited on the sensing platform [46]. The contribution of the charge-transfer (CT) mode to SERS is mainly based on the change of the molecule polarizability attached to the nanostructured surface, which can lead to new metal–analyte complex formation. Chemical enhancement is much weaker than EM enhancement [9]. Such a structure is shown in Figure 5. Titanium oxide nanotubes coated with silver nanoparticles (0.02 mg/cm^2^) smaller than 100 nm can be observed on the 3D titanium surface. Spherical nanoparticles forming agglomerates are located at the edges of the nanotubes, but also decorate the inner and outer walls because the nanotubes are separated from each [17,28]. The side view shows exactly how the silver deposit is arranged. The topography of nanotubes ensures a homogeneous distribution of silver on their surface, and at the same time generates the formation of privileged slits, cavities and gaps between Ag NPs, which affect the SERS activity.

The literature shows that types of SERS-active surfaces can be distinguished that effectively generate the SERS signal:-“With first-generation hot spots that appear as a result of interaction between a single plasmonic nano-object and incident radiation;-With second-generation hot spots that are produced by coupled plasmonic nano-objects with controllable interparticle distances;-With third-generation hot spots that are a product of superposition of electromagnetic field originating from metal NPs and electromagnetic field scattered from the backing platform [9]”.

Taking into account the above data and looking at Figure 5, it can be seen that the dominant mechanism of creating hot spots in our case may be second-generation interaction, because the proper distance between nano-objects is created by the surface topography of the nanotubes. Nanotubes form a densely packed structure and are separated from each other (see Figure 3).

A different surface topography was obtained for the 3D titanium substrate without titanium oxide nanotubes but with a silver deposit of the same amount (0.02 mg/cm^2^) (see Figure 6). In this case, we did not observe such a strong development of the nanoscale surface. Silver, just like nanotubes, precisely covers the surface of the 3D titanium substrate, and the resulting silver structure appears to be morphologically slightly roughened, without significant topographic changes. The only changes are due to microscale spherical accretions.

The obtained materials, which were microscopically characterized, were also analyzed for their chemical composition using XPS spectroscopy. Figure 7 shows XPS Ti2p high-resolution spectra for 3D titanium substrate in the as-made state (PBF-LP process), as well as after anodic oxidation and annealing at a temperature of 450 °C and silver deposition on both substrates. For all the tested samples, it can be seen that the peak position of the Ti2p_3/2_ peak corresponds to an energy of ~458.8 eV, which suggests the presence of stoichiometric titanium oxide TiO_2_ [47]. Moreover, it can be observed in the Ti2p spectrum of the Ti 3D substrate that two additional components can be distinguished at lower binding energies. The position of these additional peaks is attributed to the presence of nitrides (454.5 eV) [48,49] and metallic titanium and nonstoichiometric oxides or oxynitrides of Ti (456.6 eV) [49,50]. This is confirmed by the signal recorded from the nitrogen N1s, where, after peak deconvolution, the characteristic positions of the peak maxima for nitrides at energies 397.2 (TiN) [48,51] and 396.1 eV (TiO_x_N_y_) [49] were distinguished. The presence of nitrides in the 3D titanium initial state is related to the manufacturing process itself, where it is impossible to remove from the preparation chamber all nitrogen and oxygen and keep the process in pure argon atmosphere [52]. The construction of the Realizer SLM50 machine where the rotary pump is unable to remove all gas residues from the building chamber may favor the formation of nitrides/oxynitrides during the powder bed fusion process using a laser beam. However, for the Ti 3D sample with TiO_2_ nanotubes, after annealing at 450 °C, no signals from nitrides were observed, only signals from titanium oxides (~458.8 eV (TiO_2_), ~460.9 eV (TiO_x_)). The absence of nitrides in this case is related to the fact that the thickness of the nanoporous layer after anodizing and annealing at 450 °C is about 800 nm, which we observed in our previous work [41]. It should be noted here that the depth resolution of the XPS method is several—tens of nm depending on the type of material analyzed [53,54,55,56]. Another reason is that the Ti-N bonds decompose under the influence of temperature, which was observed during the introduction of nitrogen into the TiO_2_ structure under the influence of thermochemical treatment [56]. Similar results were obtained for the sample with the silver deposit (see Figure 8).

After the silver deposition (see Figure 8), the XPS analysis showed the presence of this element. A strong Ag3d silver signal was recorded. The deconvolution of this signal into two components contributed to assigning the main maximum to the metallic silver (~368.0 eV) and the signal of lower intensity to silver oxide (~369.0 eV) [37,57]. It should be noted here that the Ag3d silver peak for both substrates was over 96% of the sum of all other signals from the analyzed surfaces.

In the next step, the prepared substrates with silver deposit were used for SERS measurements. Figure 8 shows the spectra of adsorbed pyridine on the surface of 3D titanium subjected to surface functionalization and the sample in its initial state. The spectra show the characteristic positions of the bands of pyridine resulting from the vibrations of the aromatic ring (at about 1004 cm^−1^ due to the ν_1_ vibration, and at 1032 cm^−1^ due to the ν_12_ vibration), which are typical bands of pyridine adsorbed on the nanostructured silver [58]. Comparing the two spectra, it can be seen that the SERS intensity is several hundred times higher for the sample with TiO_2_ nanotubes than for the sample not subjected to nanofunctionalization. The characteristic distribution of nanoparticles on the surface of the nanotubes (nanorings) favors the generation of strong electromagnetic fields that amplify the signal from the adsorbed probe molecule under the influence of laser light. According to the theory of plasmon resonance, at each measurement step, the radiation is proportional to the square of the electric field gain, which yields a total gain of the SERS signal proportional to the fourth power of the field enhancement [5,9,45]. This is closely related to the formation of “hot spots”, of which there are definitely more on the laser powder bed fused 3D titanium surface covered with nanotubes and silver. Therefore, they can form second-generation hot spots produced by coupled plasmonic nano-objects with controllable interparticle distances. A lack of an appropriate nanotopography is not conducive to the generation of strong electromagnetic fields around the nanoparticles; therefore, for the sample without nanotubes, a SERS spectrum of much lower intensity was recorded (see Figure 9). Apart from the characteristic bands from pyridine, the signals that can be seen in the spectra are related to carbon species at 1100 cm^−1^ and ~1600 cm^−1^ [59].

As mentioned earlier, the distribution of silver on the nanotubes and the specific nanotopography are of key importance in enhancing the SERS signal. Figure 10 shows maps of the SERS signal distribution on the surface of the silver-decorated samples. These results clearly show the influence of the substrate on the change in the intensity of the recorded spectra. The concentration of hot spots is much greater on the powder bed fused 3D titanium surface with TiO_2_ nanotubes and silver deposit than it is on the 3D titanium without nanotubes, as can be seen on the scale showing the change in SERS signal intensity (see legend). In the case of the sample without nanotubes, it can be seen that the signal is uniform over the entire analyzed surface. On the other hand, in the case of the 3D titanium substrate with nanotubes, the signal intensity changes very clearly with the appearance of local gain maxima. This effect can be attributed to the influence of the sample topography, which is attributed to the 3D titanium substrate after the PBF-LB process (see Figure 2) and the nanotopography after surface functionalization by anodic oxidation (see Figure 4 and Figure 5).

Based on the SERS spectra collected for the materials produced with silver deposit, the enhancement factor (E_F_) for adsorbed pyridine was estimated (for the band ν_1_ at 1004 cm^−1^), based on the following formula: E_F_ = I_SERS_/I_REF_ × hC_REF_/N_SURF_, where I_SERS_ and I_REF_ are the Raman intensities obtained from the SERS and normal Raman (NR) investigations, respectively, C_REF_ is the concentration of pure pyridine in the NR measurements and h is the depth-of-focus of the laser beam. The average number of adsorbed molecules of pyridine per geometrical surface area unit participating in the SERS measurements (N_SURF_) was calculated assuming that the adsorbed molecules are spheres closely packed on a plane to form a hexagonal lattice [10,18,28]. Therefore, the calculated E_F_ for the powder bed fused 3D Ti substrate without nanotubes was 2.24 × 10^3^ and for the substrate with nanotubes 1.26 × 10^6^. This huge change in the E_F_ for both substrates is apparently due to the influence of the surface morphology. On the substrate with nanotubes, there are definitely more privileged places generating very strong electromagnetic field amplification around the silver nanoparticles. These are places such as the gaps between the nanotubes, where the appropriate distance between the silver nanoparticles is maintained, as well as the nanopores themselves, which are covered with silver [18,20,41,43]. In addition, the original geometry of the titanium sample after the PBF-LP process with characteristic spherical accretions also affects the enhancement effect, as can be seen on the map shown in Figure 10.

The results obtained were additionally compared with our previous works, in which we used titanium oxide nanotubes to design and manufacture SERS platforms. For comparative purposes, a substrate was selected where nanotubes of the same size (~80 nm) on Ti foil were decorated with silver (0.02 mg/cm^2^). Silver nanoparticles were deposited on a Ti foil nanotube substrate using the same method as in this study, using the sputtering method in a vacuum [60]. The collected results are presented in Figure 11 and compared with the reference sample of the electrochemically roughened silver electrode [28,57]. It can be seen that the TiO_2_ nanotubes formed on the 3D Ti surface have the highest enhancement factor 1.26 × 10^6^. Lower value of E_F_ for the sample with TiO_2_ nanotubes on Ti foil and with a silver deposit of 0.02 mg/cm^2^ (E_F_ = 6.83 × 10^5^) was obtained and for the electrochemically roughened Ag electrode (EF = 4.55 × 10^5^, reference platform) [28,57]. This means that the E_F_, in addition to the surface nanotopography, is influenced by the effect associated with the substrate in its original state. For flat surfaces such as Ti foil, the E_F_ is lower than for printed titanium. This comparison clearly shows that the surface topography of the printed titanium in its initial state (see Figure 2) has a significant impact on the final result related to the estimate of the E_F_ of the produced platforms, despite the similar surface nanotopography after surface functionalization. Other researchers who used printed platforms for SERS applications came to similar conclusions. They showed that printed Cu−PLA composite disks with silver film deposited by galvanic displacement proved extremely attractive for detection of environmental contaminants in water. They also noted that the combination of 3D printing with SERS measurements opens up new possibilities in the development of this method [21].

## 4. Conclusions

This work shows the possibility of the application of powder bed fusion using laser beam (PBF) technology to design and manufacture 3D titanium SERS platforms. The 3D titanium-based platforms obtained by the PBF-LB process, after proper functionalization of the nanoscale surface by anodic oxidation, had an enhancement factor of 1.26 × 10^6^ estimated for pyridine, which was used as a probe molecule in the SERS studies. The main factors influencing the final enhancement of the produced platforms were topographic effects, both in the initial state and after the nanofunctionalization of the surface. Therefore, this kind of substrate can form second-generation SERS hot spots produced by coupled plasmonic nano-objects (Ag NPs) with controllable distances. The distance is controlled by titanium oxide nanotubes produced by anodic oxidation, which form a dense structure on the surface of the printed titanium. The gaps, slits and cavities play a crucial role in enhancing the SERS signal, where the electromagnetic mechanism is the most important factor responsible for this phenomenon. Preliminary studies on the use of titanium powders in PBF-LB technology for the preparation of SERS platforms have shown that this process opens up new design possibilities, in particular in analytical chemistry and biochemistry. The only limitation of this process is a lateral resolution of the formation of the structures (microscale printing). Therefore, further surface functionalization of the materials produced in this way is required at the nanoscale, where the plasmonic effects are much more effective.

## Figures and Tables

**Figure 1 materials-15-03108-f001:**
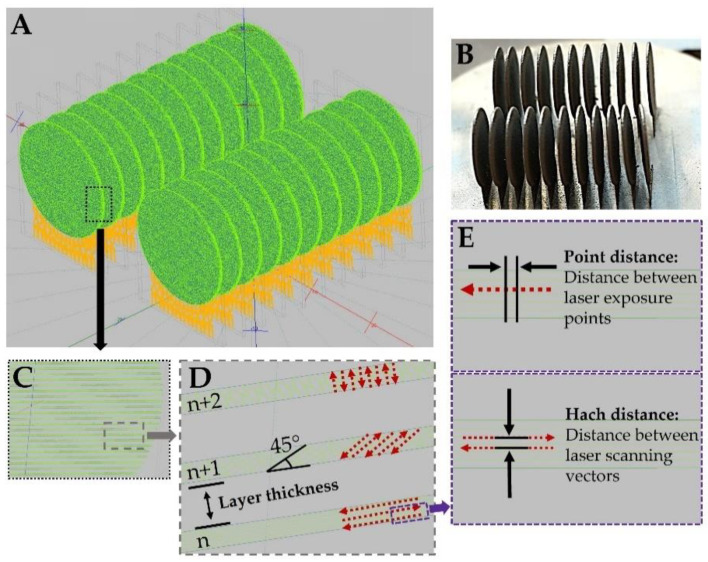
CAD models of sample discs (green) with support structures (yellow) after PBF-LB process parameters (**A**); fabricated 3D titanium samples (**B**); representative disc sample with visible layers in CAD model (**C**); alternating scanning strategy with visible distance between layers (**D**); laser exposure points (point distance) and distance between laser scanning vectors (hatch distance) (**E**).

**Figure 2 materials-15-03108-f002:**
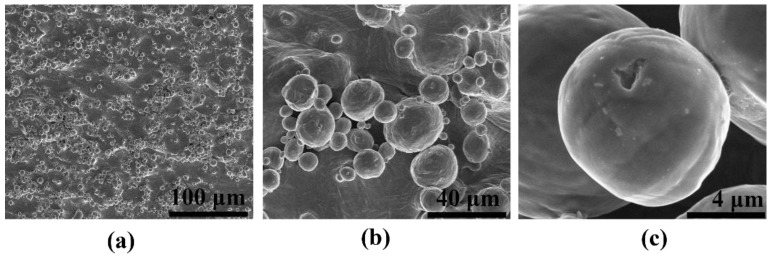
SEM images of the surface topography of the Ti 3D after PBF-LB process. Magnification of images is: (**a**) 100×, (**b**) 1000× and (**c**) 10,000×.

**Figure 3 materials-15-03108-f003:**
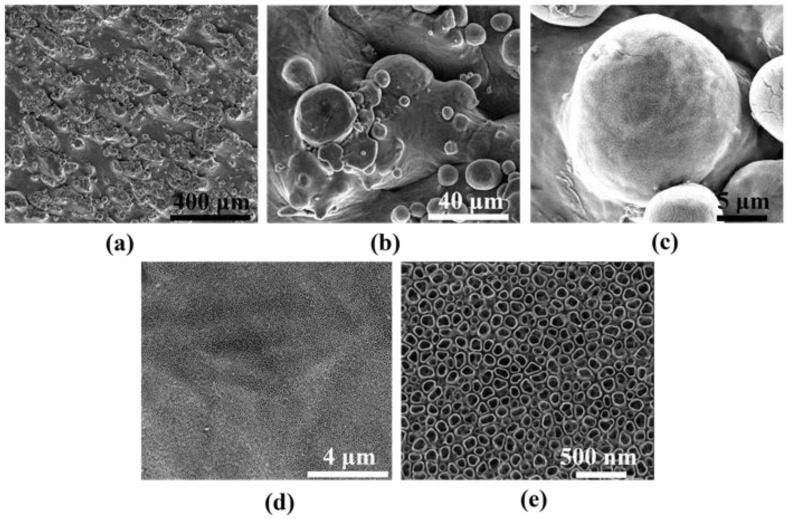
SEM images of Ti 3D surface after anodic oxidation in an electrolyte based on glycerin and water with ammonium fluoride and heating at 450 °C (low and high magnification). Magnification of images is: (**a**) 100×, (**b**) 1000×, (**c**) 5000×, (**d**) 10,000× and (**e**) 50,000×.

**Figure 4 materials-15-03108-f004:**
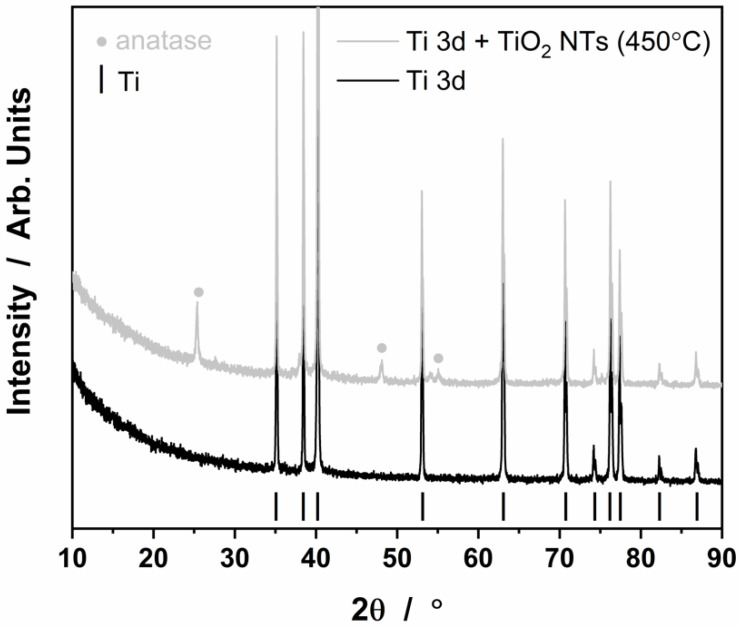
XRD patterns for the Ti 3D substrate after PBF-LB process and annealed at 450 °C.

**Figure 5 materials-15-03108-f005:**
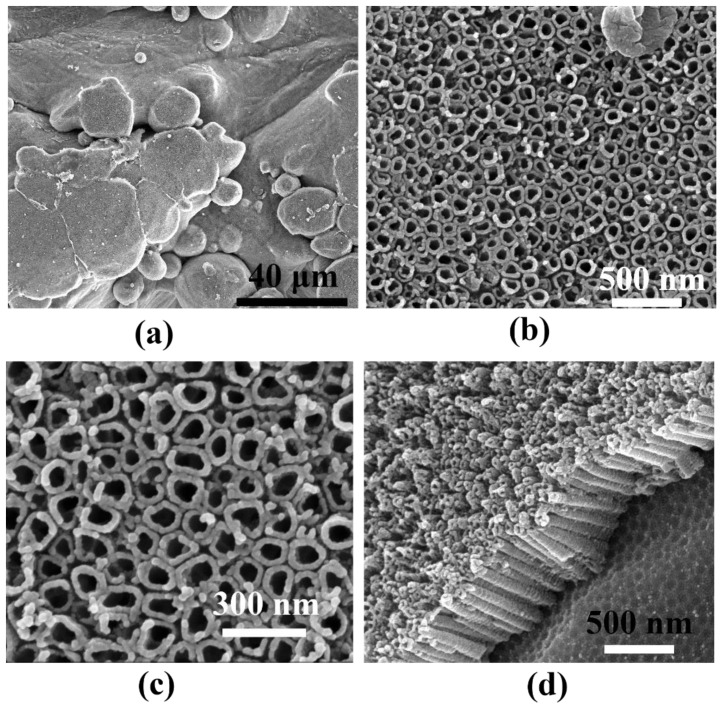
SEM images of Ti 3D surface after anodic oxidation and silver deposition by magnetron sputtering (0.02 mg/cm^2^). Magnification of images is: (**a**) 1000×, (**b**) 50,000× and (**c**) 100,000×. Side view of TiO_2_ NTs with silver deposit (**d**).

**Figure 6 materials-15-03108-f006:**
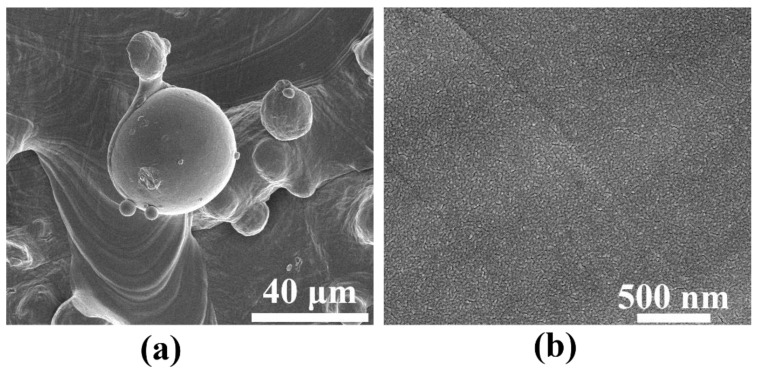
SEM images of Ti 3D surface after PBF-LB process with silver deposit (0.02 mg/cm^2^) in (**a**) low and (**b**) high magnification.

**Figure 7 materials-15-03108-f007:**
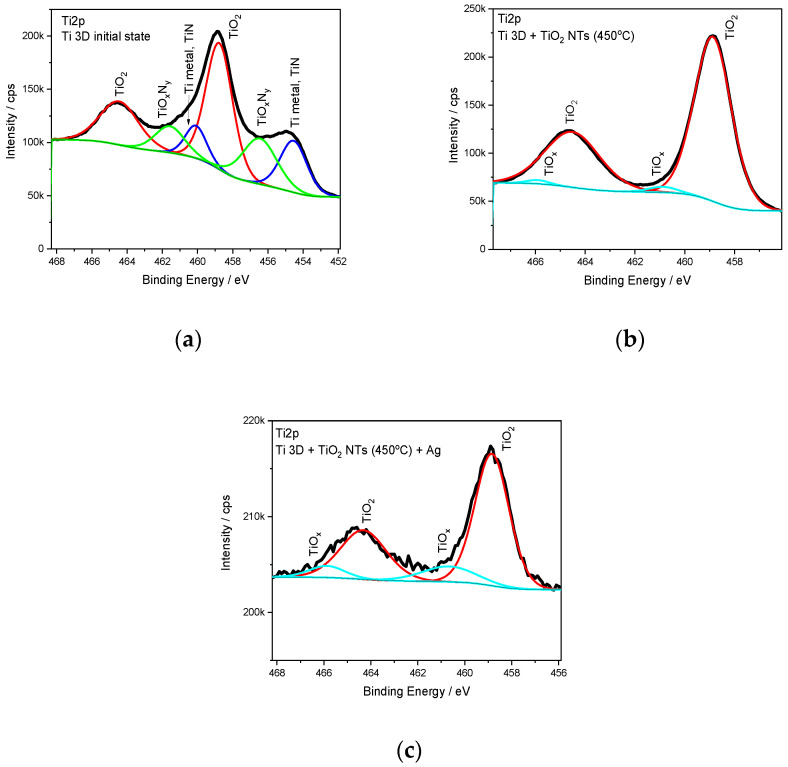
High-resolution XPS spectra of titanium Ti2p (**a**) for the Ti 3D substrate, (**b**) after anodic oxidation and annealing at 450 °C and (**c**) after silver deposition (0.02 mg/cm^2^) on both substrates.

**Figure 8 materials-15-03108-f008:**
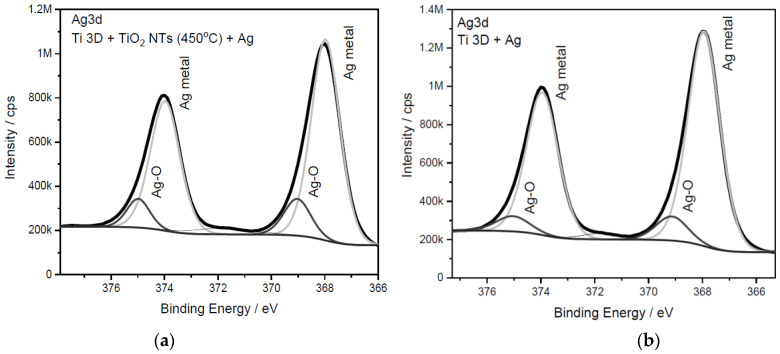
(**a**) XPS high-resolution spectra of silver Ag3d for Ti 3D substrate (**b**) after anodic oxidation process and annealing at 450 °C.

**Figure 9 materials-15-03108-f009:**
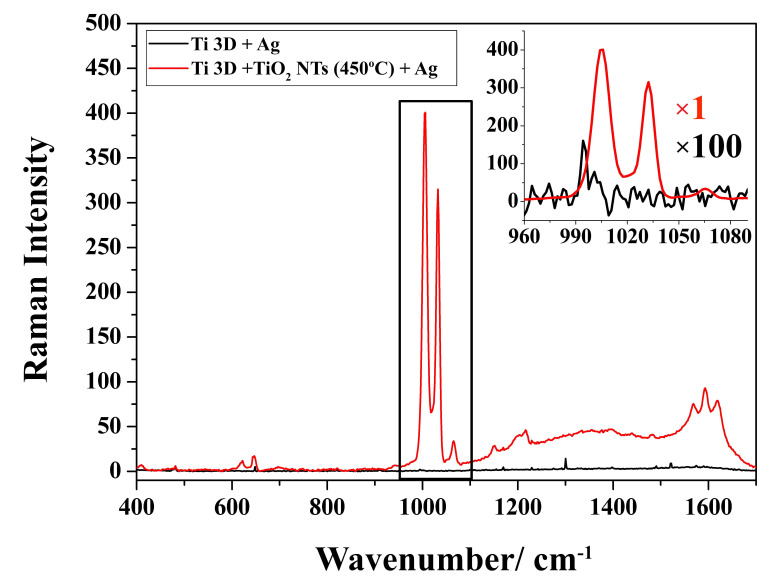
SERS spectra of adsorbed pyridine (0.05 M + 0.1 M KCl) on the surface of samples Ti 3D + TiO_2_ NTs (450 °C) + 0.02 mg/cm^2^ and Ti 3D + 0.02 mg/cm^2^.

**Figure 10 materials-15-03108-f010:**
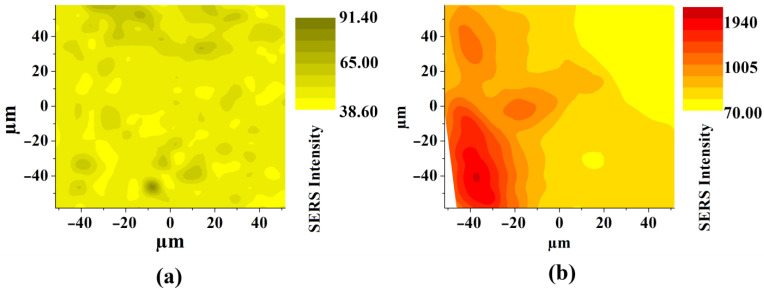
SERS spectrum distribution maps recorded on the (**a**) Ti 3D with 0.02 mg/cm^2^ silver deposit, (**b**) Ti 3D with TiO_2_ nanotubes and Ag deposit (0.02 mg/cm^2^).

**Figure 11 materials-15-03108-f011:**
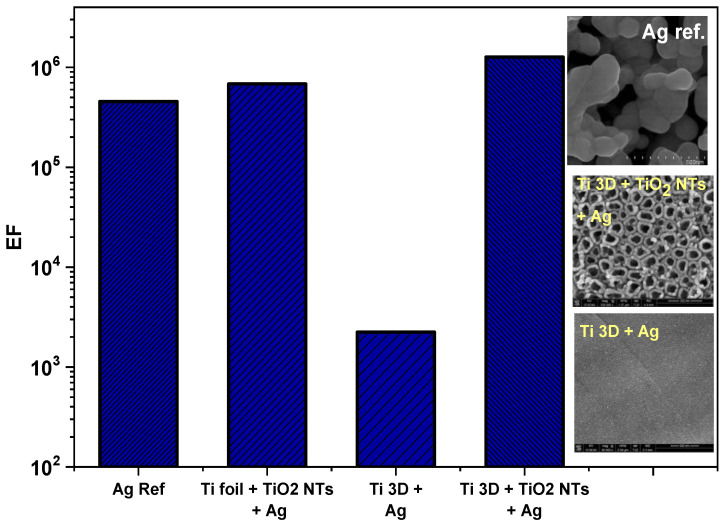
Comparison of the estimated E_F_ from adsorbed pyridine (0.05 M + 0.1 M KCl) on the fabricated platforms in this work with TiO_2_ nanotube-based substrates formed on a Ti foil with a silver deposit. An electrochemically roughened silver electrode was used as a reference. Illustrative SEM images for the discussed samples are attached to the Figure.

## Data Availability

Not applicable.

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
