# Peer review of "Nanofunctionalization of Additively Manufactured Titanium Substrates for Surface-Enhanced Raman Spectroscopy Measurements"

_materials, 2022, doi:10.3390/ma15093108_

Round 1

Reviewer 1 Report

The paper “Nano-Functionalization of additively manufactured titanium substrates for SERS measurements” investigates the use of laser powder bed fusion process made it possible to generate an original geometry of the SERS substrate, which was further functionalized at the nanoscale. Some suggestions:

  • A graphical abstract would add interest to catch the eye
  • In the abstract introduce quantifiable observed results.
  • The end of the introduction should state the novelty of the paper presented.
  • 9 is too blurry.
  • The conclusions are more on the side of the technical summary

Author Response

First, the Authors would like to thank the Reviewer for valuable comments on the work. We modified our work according to all your suggestions and all changes introduced to the text were marked in red.

The paper “Nano-Functionalization of additively manufactured titanium substrates for SERS measurements” investigates the use of laser powder bed fusion process made it possible to generate an original geometry of the SERS substrate, which was further functionalized at the nanoscale. Some suggestions:

  1. A graphical abstract would add interest to catch the eye. A graphical abstract was prepared.
  2. In the abstract introduce quantifiable observed results. Appropriate changes have been made to the abstract.
  3. The end of the introduction should state the novelty of the paper presented. Appropriate changes were made in the final part of the introduction in accordance with the comments of the Reviewer.
  4. 9 is too blurry. The Figures quality has been improved.
  5. The conclusions are more on the side of the technical summary. The conclusions of the work were corrected.

Author Response

On the beginning we would like to thank Reviewer for the remarks that allowed us to improve the quality of our work. All suggestions were taken into account. All changes in the text are marked in red.

1- Abstract does not show clearly the purpose of research. Appropriate changes have been made to the abstract, where underlined the purpose of the research.  

2- Introduction should be re-written. For instance, Line 27 surfaces appropriately prepared for this purpose, Purpose should be clarified. The introduction at this point has been corrected.

„Usually these are nanostructured substrates with a high degree of surface development based on plasmonic metals [1]”.

3- Line 89: the samples were rinsed with deionized water (24 h). Why does the rinsing take 24 hours! An appropriate explanation has been added to the text.

After anodization, the samples were cleaned with deionized water through long-term rinsing (24 h) and subsequently dried in air. Such a procedure leads to the cleaning of the sample surface from some organic contaminants coming from the electrolyte.

4- It would be better that authors show a cross section of anodised nanotubes or a tilted image of SEM surface to show the nanotubes in thickness. Thank you for this comment. The side view of the nanotubes after silver deposition has been added to Figure 5.

5- Authors should elaborate more the novelty of this paper in compare to author’s previous published studies in introduction. Appropriate changes were made in the final part of the introduction in accordance with the comments of the Reviewer.

6- There are many measurements are missing for example EDS analysis, XRD, specification of as received titanium such as particles size and distribution. However, this result can be found in published author’s paper. Then this study cannot be consider as a full paper. It is a communication or letter. Thank you for this critical remark. New elements for the discussion of the results were introduced to the work. The XRD spectra of the produced materials was added to define the structure (see Figure 4). In addition, new references of other researchers have also been added to deepen the discussion. EDS measurements for the produced materials would not contribute anything significant to the work, apart from the qualitative and quantitative results. The estimated amount of silver using the EDS method is not important in this case, the surface chemistry is more important. That is why we decided to use XPS measurements, where in addition to the chemical composition, the chemical state of the elements can be determined.

7- Scale bars in all the SEM images are not readable. The scale has been corrected in all SEM images.

8- I would suggest an EDS map on the surface of Fig 4, image b to see Ag deposition on Ti nanotubes. Ag deposition on the Ti nanotubes shows very selective and just small part covered by Ag. Why? As mentioned earlier, the EDS method is not suitable for testing this type of material. The surface plays a key role in SERS spectroscopy, which is why XPS information is so important in this case. Using XPS spectroscopy, we obtain qualitative and quantitative information as well as information on the chemical state. Moreover, the thickness of the silver deposit on the surface of nanotubes is about 20 nm (0.02 mg/cm2), so the EDS signal from Ag in relation to titanium and oxygen will be relatively small (depth resolution of EDS method is about 1 µm). Please also note that the distribution of silver on the tested surfaces is quite homogeneous as shown by the SEM images of the surface and the side view. The dominant signal in the XPS measurements (due to the depth resolution of the method) is the silver peak. In the case of XPS measurement using a non-monochromatic source, the signal is recorded from an area of 0.2 cm2. Therefore, a relatively large sample area is analyzed, which gives an average result with characteristic depth resolution for XPS.

9- Line 344: In conclusion, authors mentioned, the main factors influencing the final enhancement of the produced platforms were topographic effects, both in the initial state and after the nano-functionalisation of the surface. I did not find any strong evidence for this. Taking into account the extended description of the research results, the conclusions was changed in line with the Reviewer's comment.

Reviewer 3 Report

This work shows the possibility of using LPBF technology for the production of 3D titanium substrates for surface-enhanced Raman scattering (SERS) measurements. The contents of current work should be of interest for the field. However, the following comments should be addressed.

  • In the Abstract, it writes “using LPBF technology by selective laser melting (SLM) process”, LPBF and SLM are repetitive, and it is enough to mention LPBF in this manuscript. In addition, “laser powder bed fusion method” in the Introduction/Conclusions and “Laser Powder Bed Fusion” in Section 2.1 should be unified.
  • In Section 2, many parameters and equipment are introduced, which is not that we focus on. Add more analysis in this Section.
  • On Page 9, “” and “EF” should be unified.
  • The entire work is morelike a review article, in the Conclusions, the main points of current work or innovations should be highlighted, not just a simple summary.

Author Response

Thank you for critical comments on the work, all comments were taken into account and the text has been appropriately changed. All changes in the text are marked in red.

This work shows the possibility of using LPBF technology for the production of 3D titanium substrates for surface-enhanced Raman scattering (SERS) measurements. The contents of current work should be of interest for the field. However, the following comments should be addressed.

  1. In the Abstract, it writes “using LPBF technology by selective laser melting (SLM) process”, LPBF and SLM are repetitive, and it is enough to mention LPBF in this manuscript. In addition, “laser powder bed fusion method” in the Introduction/Conclusions and “Laser Powder Bed Fusion” in Section 2.1 should be unified. The description was unified.
  2. In Section 2, many parameters and equipment are introduced, which is not that we focus on. Add more analysis in this Section. The description of point 2.1 has been corrected.
  3. On Page 9, “” and “EF” should be unified. The description was unified.
  4. The entire work is more like a review article, in the Conclusions, the main points of current work or innovations should be highlighted, not just a simple summary. As a result of the changes introduced to the discussion of the results, their summary was modified, in particular, special attention was paid to emphasizing the novelty of the work.

Reviewer 4 Report

The paper contains some interesting results that make it publishable in the journal after the following mandatory revisions:

1-avoid using abbreviated words on the title of the paper.

2-in the abstract, the authors should mention what parameters were investigated in this research. Also, the tests used and the general results should be briefly mentioned in this section.

3-Some old references were used in the introduction of the paper which is not acceptable. Also the introduction is very short. On the other hand, the correlation between the surface coatings, and surface modification on the properties were not discussed and analyzed. The current introduction is incomplete and the following documents can be used to strengthen it:

- Deposition of ceramic nanocomposite coatings by electroplating process: a review of layer-deposition mechanisms and effective parameters on the formation of the coating, Ceramics International, Vol. 45 (17), 2019, 21835-21842.

- CLAD ALUMINUM ALLOY PRODUCTS AND METHODS OF MAKING THE SAME, US Patent 20,180,304,584.

- Facile Synthesis of Ag Nanowire/TiO2 and Ag Nanowire/TiO2/GO Nanocomposites for Photocatalytic Degradation of Rhodamine B, Materials, Vol. 14 (4), 2021, 763.

4-figure captions are long. Captions should be as short as possible.

5-The texture morphology obtained in figure 2 should be explained. What was the reason for the spherical morphology?

6-The scales of the microstructural images of figures 3 and 4 are not clear.

7-The conclusions are better to be as bullet points.

Author Response

Thank you for all comments on the work, which were included in the next version of the manuscript. All changes in the text are marked in red.

The paper contains some interesting results that make it publishable in the journal after the following mandatory revisions:

1-avoid using abbreviated words on the title of the paper. The title of the work has been corrected.

2-in the abstract, the authors should mention what parameters were investigated in this research. Also, the tests used and the general results should be briefly mentioned in this section. Appropriate changes have been made to the abstract.

3-Some old references were used in the introduction of the paper which is not acceptable. Also the introduction is very short. On the other hand, the correlation between the surface coatings, and surface modification on the properties were not discussed and analyzed. The current introduction is incomplete and the following documents can be used to strengthen it:

We agree with the Reviewer's opinion additional references have been introduced into the work.

- Deposition of ceramic nanocomposite coatings by electroplating process: a review of layer-deposition mechanisms and effective parameters on the formation of the coating, Ceramics International, Vol. 45 (17), 2019, 21835-21842.

- Facile Synthesis of Ag Nanowire/TiO2 and Ag Nanowire/TiO2/GO Nanocomposites for Photocatalytic Degradation of Rhodamine B, Materials, Vol. 14 (4), 2021, 763.

4-figure captions are long. Captions should be as short as possible. Where it was possible the figure captions were modified.

5-The texture morphology obtained in figure 2 should be explained. What was the reason for the spherical morphology? An appropriate comment has been added.

One of the challenges in LPBF processing of the metal parts are loose, unmelted powder particles fused to all external surfaces of fabricated parts. This phenomenon is explained by the nature of the melt pool in LPBF fabrication. The high-energy laser beam produces sufficient heat to create the melt pool at the surface of the powder bed. The temperature surrounding the melt pool is immediately increased by heat conduction. Some of the powder particles, regardless the CAD design, which are placed in the heat conduction zone do not melt but may become lightly sintered to the surface of the part. In our study,  the unmelted powder particles on the disc's surfaces are caused also by heat transfer, between the melt pool and unmelted powder in the bed, affecting their sintering with the surface. The size of unmelted fully powder particles visible on the surface discs is within the range of the used raw powders and is from a few to several dozen micrometers. Slightly sintered titanium particles remain on the 3D Ti surface even after washing in an ultrasonic cleaner.  The appearance of these non-melted Ti particles is associated with the completion of the individual disc printing process which leads to raw surface (see Fig. 1).

6-The scales of the microstructural images of figures 3 and 4 are not clear. The scale has been corrected in all SEM images.

7-The conclusions are better to be as bullet points. Taking into account the extended description of the research results, the conclusions was changed.

Round 2

Reviewer 1 Report

The paper has improved significantly. The work seems interesting and publishable. I would like to thank the editor for allowing me to review this type of work and the authors for their research efforts.

Author Response

The authors would like to thank the Reviewer for valuable comments on the work and its approval.

Reviewer 4 Report

The requested revisions are applied. The paper can be published in its current format.

Author Response

The authors thank the Reviewer for comments regarding to the manuscript and its acceptance.